# Measurement of Electron Density from Stark-Broadened Spectral Lines Appearing in Silver Nanomaterial Plasma

**Ashraf M. EL Sherbini [1]** [ID]**, Ahmed E. EL Sherbini [1] and Christian G. Parigger [2],*** [ID]

[1] Laboratory of Laser and New Materials, Faculty of Science, Cairo University, Giza 12613, Egypt; elsherbiniaa@gmail.com (A.M.EL.S.); ahmed.azdakyne@gmail.com (A.E.EL.S.)

[2] Department of Physics and Astronomy, University of Tennessee/University of Tennessee Space Institute, 411 B. H. Goethert Parkway, Tullahoma, TN 37388, USA

* Correspondence: cparigge@tennessee.edu; Tel.: +1-931-841-5690

**Abstract:** This work communicates results from optical emission spectroscopy following laser-induced optical breakdown at or near nanomaterial. Selected atomic lines of silver are evaluated for a consistent determination of electron density. Comparisons are presented with Balmer series hydrogen results. Measurements free of self-absorption effects are of particular interest. For several silver lines, asymmetries are observed in the recorded line profiles. Electron densities of interest range from 0.5 to $3 \times 10^{17}$ cm$^{-3}$ for five nanosecond Q-switched Nd:YAG radiation at wavelengths of 1064 nm, 532 nm, and 355 nm and for selected silver emission lines including 328.06 nm, 338.28 nm, 768.7 nm, and 827.3 nm and the hydrogen alpha Balmer series line at 656.3 nm. Line asymmetries are presented for the 328.06-nm and 338.28-nm Ag I lines that are measured following generation of the plasma due to multiple photon absorption. This work explores electron density variations for different irradiance levels and reports spectral line asymmetry of resonance lines for different laser fluence levels.

**Keywords:** laser-induced plasma; Stark broadening; electron density; nanomaterial; atomic spectroscopy; silver; hydrogen

## 1. Introduction

Laser-induced breakdown spectroscopy (LIBS) [1] is utilized for measuring plasma that is generated at or near silver nanomaterial. During the last few decades, LIBS has been recognized for its versatility and integral aspect for a variety of spectro-chemical analysis procedures. Typically, high peak power and nominal nanosecond radiation is focused to irradiance levels of the order of a few MW/cm$^2$ to TW/cm$^2$ and the emitted light is recorded with spectrometers and gated array detectors [2,3]. Historically, laser-induced plasma spectroscopy (LIPS) explores the physics of the plasma induced by laser light via optical emission spectroscopy (OES) [4–7]. Spectral line shape analysis via OES leads to the determination of at least one characteristic plasma parameter such as electron density, $n_e$.

The measurement of electron density is of prime importance for the description of the plasma induced by laser radiation. Spectroscopically, $n_e$ can be measured using different experimental techniques that include the measurement of the optical refractivity of the plasma [4–7], calculation of the principal quantum number in the series limit [4–7], measurement of the absolute emission coefficient (spectral radiance per wavelength in W sr$^{-1}$ m$^{-3}$) of a spectral line [8], and measurement of the absolute emissivity of the continuum emission (W sr$^{-1}$ m$^{-3}$) [8]. However, measurement of Stark broadening of emitted lines for $n_e$ determination has been widely utilized [4–8].

Measurements of $n_e$ from Stark broadening is relatively straightforward provided that the Stark effect is the dominant broadening mechanism with significantly smaller contributions from Doppler broadening and other pressure broadening mechanisms resulting from collisions with neutral atoms (i.e., resonance and Van der Waals broadening) [4–7]. Theoretical calculations of Stark broadening parameters of hydrogen and hydrogenic lines are communicated by H. Griem [4] and E. Oks [9–11]. Precise fitting of the measured line shapes to convolutions of Lorentzian and Gaussian spectral line shapes (i.e., Voigt line shapes) allows one to extract the Stark full width at half maximum (FWHM). Subsequently, the electron density can be inferred from tabulated Stark broadening tables.

## 2. Nanomaterial

Nanomaterials usually describe structured components with at least one dimension less than 100 nm [12]. Two principal factors cause the properties of nanomaterials to differ significantly from bulk materials such as the increase in the relative surface area and the quantum effects. These factors can substantially change and/or enhance the well-known bulk properties such as chemical reactivity [13], mechanical strength [14], electrical and magnetic [15], and optical characteristics [16]. As the particle size decreases, a greater proportion of atoms are found at the surface than in the interior [17]. The quantum effects can begin to dominate the properties of matter as its size is reduced to the nano-scale. Nanoparticles are of interest because of their inherent new properties when compared with larger particles of the same materials [12–17].

It was found that the addition of a thin layer of gold and silver nanoparticles to the surface of an analyte matrix alloys can lead to signal improvement and, therefore, an improved limit of detection (LOD) in LIBS applications [18]. The acronym associated with improved emission signals is Nano-Enhanced Laser Induced Breakdown Spectroscopy (NELIBS). Conversely, interaction of high peak power radiation with pure nanomaterial targets [19–22] is investigated with so-called Nano-Enhanced Laser Induced Plasma spectroscopy (NELIPS).

In previous NELIPS work, the signal enhancement shows the following trends: (1) the enhanced emission from the nanomaterials increases linearly with time delays when compared with bulk material [19], (2) the enhanced emission increases with decreasing laser fluence [20], (3) there are no apparent changes of the plasma electron density and temperature [21,22], (4) the enhancement factors that may vary for different experimental conditions can be associated with the relative masses ejected from the targets [21,22], (5) the threshold of the plasma ignition from the surface of the nanomaterials is much smaller than that from the corresponding bulk [21,22], (6) the breakdown threshold is inversely proportional to the square of the incident laser wavelength [20–22], and, lastly, (7) the threshold of the plasma from the nano-material targets changes linearly with the diameter size of the nanoparticles [22].

Moreover, the modeling of the laser induced plasma from either type of targets (Bulk and Nano) has been theoretically investigated after the addition of a laser wavelength dependent term [21,22], which was found to contribute by 90% when using near UV laser wavelengths [21,22].

## 3. Materials and Methods

This work utilizes the same experimental setup reported in previous studies [21,22]. It is comprised of an Nd-YAG laser device (type Quantel-Brilliant B, France [19–22]) operated at the fundamental wavelength of 1064 nm and two higher harmonics at 532 nm and 355 nm with output laser energy of $30 \pm 3$ mJ, $100 \pm 4$ mJ, and $370 \pm 5$ mJ, respectively. The focusing lens was located at a distance about $95 \pm 1$ mm, which is away from the target material. Using a special thermal paper (supplied by Quantel®, France [19–22]), a circular laser beam spot revealed a radius of $0.27 \pm 0.03$ mm. In order to avoid laser focusing-lens chromatic aberrations, the plasma initiation was first observed in laboratory air and, subsequently, the target was displaced closer to the 100 mm focal-length achromatic lens. This routine would indicate that the plasma emission originated from the target rather than from ambient air surrounding the target. The light from the plasmas was collected using a 400 μm diameter optical fiber (with numerical aperture NA = 0.22) to the entrance slit of the SE200-Echelle

type spectrograph (Catalina Scientific, Tucson, AZ, USA [19–22]) with optical resolution of 0.02 nm per pixel with an average instrumental bandwidth of 0.2 nm. The optical fiber was positioned at a distance of 5 mm from the laser-plasma axis with a precise xyz-holder. The resolved spectra were monitored using a fast response intensified charge-coupled device (ICCD) (type Andor-iStar DH734-18F, Belfast, Northern Ireland [19–22]) and the data acquisition was carried out using KestrelSpec® 3.96 software Catalina Scientific, Tucson, AZ, USA [19–22]) at a resolution of 0.02 nm per pixel (of size 196 μm²).

The nano silver was supplied as a powder (MKNano®, Toronto, ON, CA [19–22]) with the product label MKN-Ag-090, CAS Number 7440-22-4, and with an average size of 90 ± 10 nm. The nano-powder is compressed to circular disk tablets with a diameter of 10 mm using a 500 kg/cm² mechanical press. The shape of the nanoparticles was investigated with transmission electron microscopy (TEM) after compression. Nearly spherical diameters of 95 ± 15 nm are found and only slight distortions were observed. Both delay and gate times were adjusted to the levels of 2 μs across the experimental studies. Background stray light during experimental runs was measured and subtracted with the help of Andor iStar ICCD-KestrelSpec® software (Catalina Scientific, Tucson, AZ, USA [19–22]). The noise level from the detection electronics was recorded across the entire wavelength region (250–850 nm) and was found to be about 20 ± 7 counts. The signal-to-noise ratio was computed using noise-level as the sum of the electronic noise in addition to the continuum emission (sometimes called background radiation) that occurs underneath the atomic lines of interest. The incident laser energy for each laser pulse was measured utilizing a quartz beam splitter. The reflected part (4%) was incident on an absolutely calibrated power-meter (Ophier model 1z02165, North Logan, UT, USA [19–22]). The laser pulse shape was measured using a 25 ps, fast-response photodiode in conjunction with a digital storage oscilloscope (Tektronix model TDS-1012, Tektronix, Beaverton, OR, USA [19–22]) and the pulse-width was found stable at a level of 5 ± 1 ns. The laser energy was adjusted with a set of calibrated neutral density filters. The absolute sensitivity of the spectrograph, camera, and optical fiber was calibrated using a DH2000-CAL lamp (supplied by Ocean Optics-SN: 037990037, Ostfildern, DE [19–22]). The data presented in this paper are taken as the average over three consecutive shots onto fresh targets and the data are presented together with standard deviations about means and plotted as error bars associated with the measurement points. The observed spectral Ag I lines (e.g., 327.9 nm, 338.2 nm, 405.5 nm, 421.2 nm, 447.6 nm, 467.7 nm, 520.9 nm, 546.5 nm, 768.7 nm, and 827.3 nm) were examined in view of self-absorption and/or self-reversal.

## 4. Results

The spectral emission is recorded in different spectral regions from the plasma that is generated by the interaction of high peak power laser radiation using the third harmonic, blue wavelength of 355 nm with silver nano-based targets and the corresponding bulk material. Figure 1 displays the measured data.

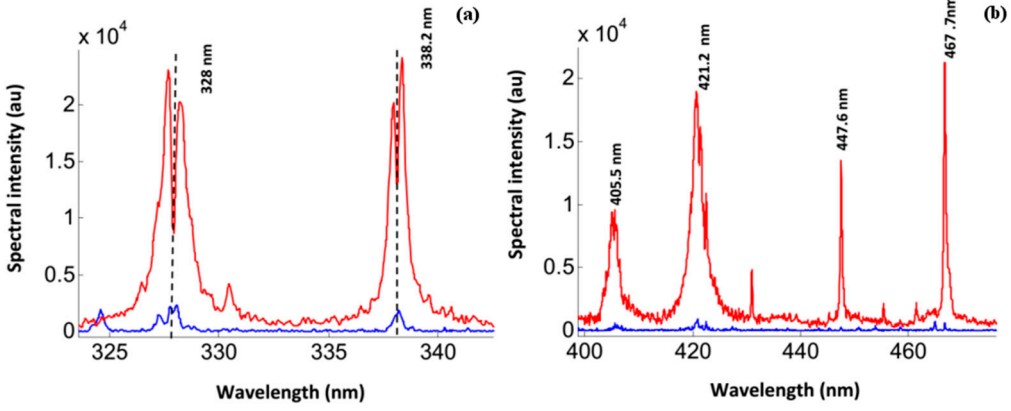

**Figure 1.** *Cont.*

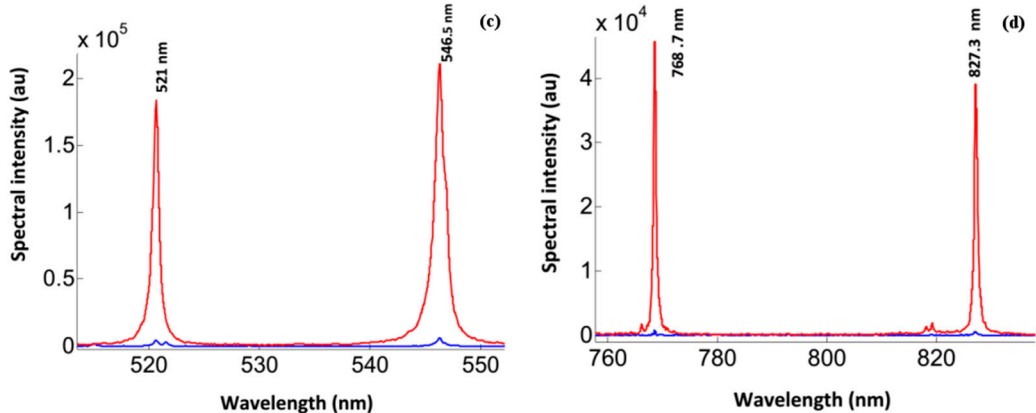

**Figure 1.** (**a**–**d**) Emission spectra at different wavelength regions from nano-based silver target (in red) and bulk target plasma (in blue).

There is a clear larger emission from the nano-based silver plasma than from plasma created from the bulk target (ratio of the red to blue curves). Detailed inspection of the resonant transitions ($4d^{10}5p$-$4d^{10}5s$) at wavelengths of 327.9 nm and 338.2 nm (as depicted in Figure 2) indicates the existence of self-reversal as well as self-absorption.

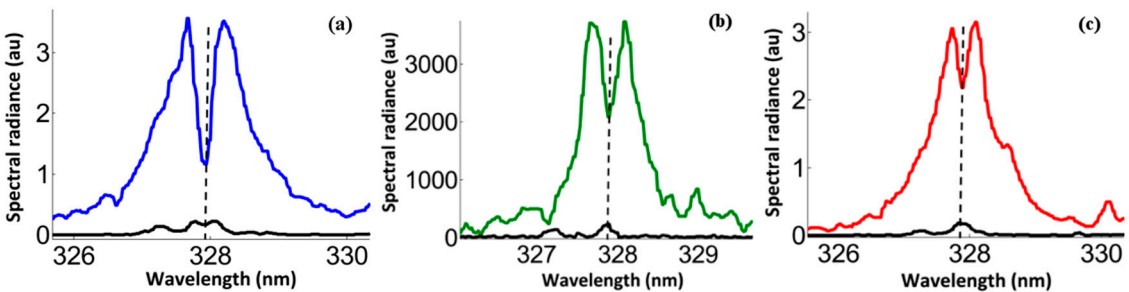

**Figure 2.** Self-reversal of the resonance Ag I line at 327.9 nm. Laser wavelengths: (**a**) 355 nm, (**b**) 532 nm, and (**c**) 1064 nm. The colored, larger, black, and lower plasma spectra are measured from nano-based and bulk silver targets, respectively.

Self-reversal is often associated with the population density of the ground state of the silver atoms ($4d^{10}5s$-state). The ground-state population exhibits a strong gradient of the plasma parameters (electron density and temperature) ranging from the plasma core to the periphery [1–5]. Moreover, this effect was found to be pronounced at a shorter wavelength laser irradiation of 355 nm (see Figure 2a—blue curve). This is in contrast with the emission from the bulk-based silver target under similar conditions.

The results are consistent with previous studies [21,22] that discussed that the population density of the ground state is larger for the plasma created at the surface of the nano-based target than for the bulk-based plasma, which is described by the enhancement factor defined by $\left( Enh.Factor = \frac{I_o^{Nano}}{I_o^{Bulk}} \approx \frac{N_o^{Nano}}{N_o^{Bulk}} \right)$. In this scenario, $\frac{I_o^{Nano}}{I_o^{Bulk}}$ is the ratio of the spectral radiance of nano-based to bulk-based target plasma spectral line. However, $\frac{I_o^{Nano}}{I_o^{Bulk}}$ is the ratio between the corresponding population density of the ground state of the silver atoms. The quantification of self-absorption and/or self-reversal that one should rely on certain optically thin (standard) spectral lines should allow precise measurements of plasma electron density [23]. The presence of $H_\alpha$ emission spectra provides a good candidate for the measurement of the plasma electron density, but $H_\alpha$ is often absent during the interaction of both green and blue lasers. Therefore, one should consider other optically thin lines that can be measured during the interaction of the different Nd:YAG harmonics with nano-based targets.

The Ag I lines at wavelengths of 768.7 nm and 827.35 nm are candidates for electron density measurements. At the reference density of $1 \times 10^{17}$ cm$^{-3}$, the Stark broadening parameters of $\omega_s^{827.25} = 0.18$ nm and $\omega_s^{768.7} = 0.17$ nm [24] allow one to estimate electron density from the Lorentzian FWHM component of each line. The fitting of the line shapes to Voigt profiles is shown in Figure 3. The overall results are summarized in Table 1 including electron densities measured from the optically thin H$_\alpha$. In Table 1, the accuracies of $n_e$ are 20%. In this work, the $n_e$ measurement from H$_\alpha$ is preferred since H$_\beta$ shows interference from other spectral features. Empirical formulae [25] are utilized for $n_e$ determination from H$_\alpha$.

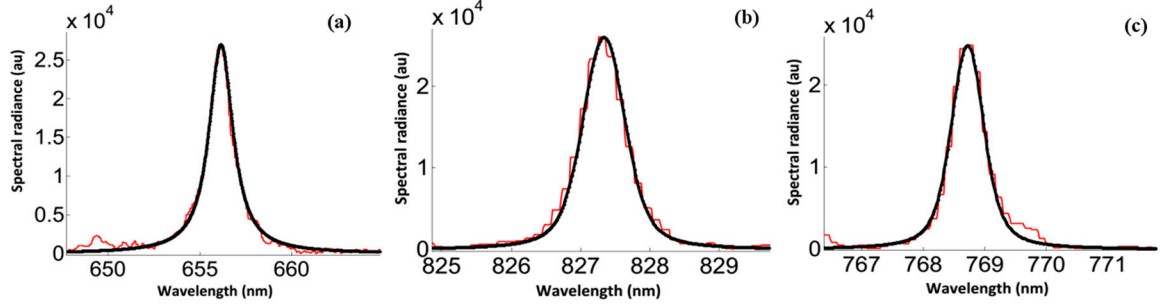

**Figure 3.** Fitting of line shapes (red) to Voigt line shape (black) for the H$_\alpha$-line (**a**) and the Ag I lines at 827.35 (**b**) and 768.7 nm (**c**) at fixed laser fluence 9.6 J/cm$^2$ and IR laser at 1064 nm excitation.

**Table 1.** Electron densities inferred from different spectral lines.

| Laser Fluence (J/cm$^2$) | $n_e$ (H$_\alpha$) ($10^{17}$ cm$^{-3}$) | $n_e^{827.35}$ Ag I: 827.35 nm ($10^{17}$ cm$^{-3}$) | $n_e^{768.9}$ Ag I: 768.7 nm ($10^{17}$ cm$^{-3}$) |
|---|---|---|---|
| 9.9 | 1.64 | 1.66 | 1.76 |
| 7.5 | 0.76 | 0.77 | 0.76 |
| 5.9 | 0.63 | 0.66 | 0.7 |
| 4.5 | 0.57 | 0.55 | 0.58 |

The results in Table 1 attest that the two lines at wavelengths of 768.7 nm and 827.35 nm can be utilized for reliable measurements of plasma electron density at the surface of silver nano-based targets during the interaction with the blue Nd:YAG laser radiation.

This work also explores optical depths of plasma created by a laser at the green and blue wavelengths. In general, the laser produced plasma is inhomogeneous even though the laser device operates in the lowest order transverse electromagnetic mode, i.e., TEM$_{00}$ [1–10]. Actually, there are two regions in the plasma produced by laser radiation. The first region includes the central hot core with a relatively large electron temperature, density, and large population densities of higher emitting species. The second region includes the outer periphery region at which the plasma becomes relatively cold (losses of internal energy by adiabatic expansion against surrounding medium) [26,27]. The second region contains a large population in lower excitation states. This situation enhances the chance or probability for some generated photons at the central region to be re-absorbed by the cold atomic species at the peripheries [26,27].

The re-absorption processes act differently over the spectral line shape. The effect at the central upshifted spectral line with little effect at the line wings is called self-reversal [28–30] and it produces a dip, which is indicated in Figure 4. The other re-absorption process is self-absorption. It is difficult to assess the level of self-absorption in n$_e$ determinations but usually larger electron densities are found from self-absorbed lines. Comparisons with well-established, optically thin emission lines can be utilized to evaluate the level of self-absorption. The existence of certain, reliable, optically thin lines can provide a measure from which one can deduce if a recorded line is optically thin or thick.

Therefore, the standard $H_\alpha$-line or other optically thin lines like the Ag I lines at 768.7 nm or 827.35 nm become important since one can use either line to check the value of the electron density.

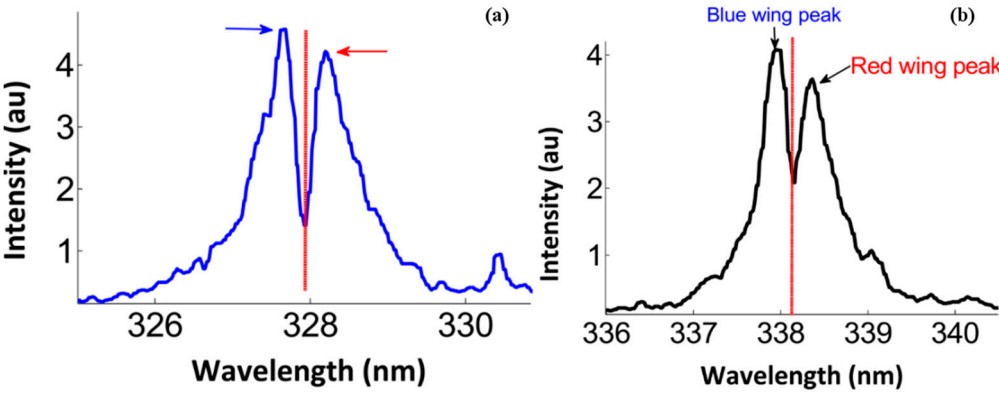

**Figure 4.** Spectral dips and asymmetries at a fluence of 9.6 J/cm$^2$. (**a**) 327.9 nm and (**b**) 338.2 nm.

Comparisons of the electron densities inferred from questionable lines with $n_e$ obtained from optically thin lines can yield the self-absorption parameter, *SA*, by using the following formula $SA \simeq \left( \frac{1-\exp(-\tau_{SA})}{\tau_{SA}} \right)$ [23]. In this scenario, $\tau_{SA} \simeq \int_{-\ell}^{0} \kappa_{SA}(\lambda) \; d\lambda$ is the plasma optical depth due to the self-absorption coefficient $\kappa_{SA}(\lambda)$ integrated over the whole spectral line region, $\Delta\lambda$, along with the line-of-sight for plasma having an approximate length $\ell$. Self-absorption causes line-shape distortions including the appearance of broader profiles with smaller spectral radiances than for optically thin lines. It is pointed out in Reference [31] that a correction to a spectral line shape against self-absorption can be carried out using a practical relation, $SA \simeq \left( n_e^{line}/n_e^* \right)^{-1.785}$, where $n_e^*$ is the electron density deduced from largely optically thin lines, e.g., $H_\alpha$ at 656.3 nm, Ag I at 768.7 or 827.35 nm, which is mentioned before. As the *SA* parameter approaches unity, the line can be considered as optically thin. In other words, the *SA* parameter determines the degree of the plasma opacity for selected spectral lines.

Similarly, the self-reversal effect can be quantified with coefficient $SR \simeq \left( \frac{1-\exp(-\tau_{SR})}{\tau_{SR}} \right)$ that can be related to transmittance $T(\tau_{SR}) \approx \left( 1/\sqrt{\pi\tau_{SR}} \right)$ [28–30] for plasma of optical depth $\tau_{SR}$. The *SR* parameter is typically not much smaller than one, but it can cause a characteristic dip at the center of the line. However, *SR* has little effect in the line wings [30]. For example, Figure 4a,b display dips for the resonance lines Ag I at 327.9 nm and 338.2 nm.

Figure 4a,b display self-reversed Ag I resonance lines at wavelengths of 327.9 nm and 338.2 nm. The spectra are recorded when using 355-nm wavelength and 9.6 J/cm$^2$ fluence excitation. Two major features occur: (i) the dip at each of the lines centers and (ii) the pronounced line asymmetry of both lines.

Notice the asymmetries in the line shapes that appear to affect both lines. The theory that can describe the spectral dip due to self-reversal does not contain an asymmetry. However, the apparent asymmetry will be subjected to further investigation in a separate study. Figure 5 displays spectral line shapes of the 327.9-nm resonance line upon 355-nm irradiation for varying levels of fluence.

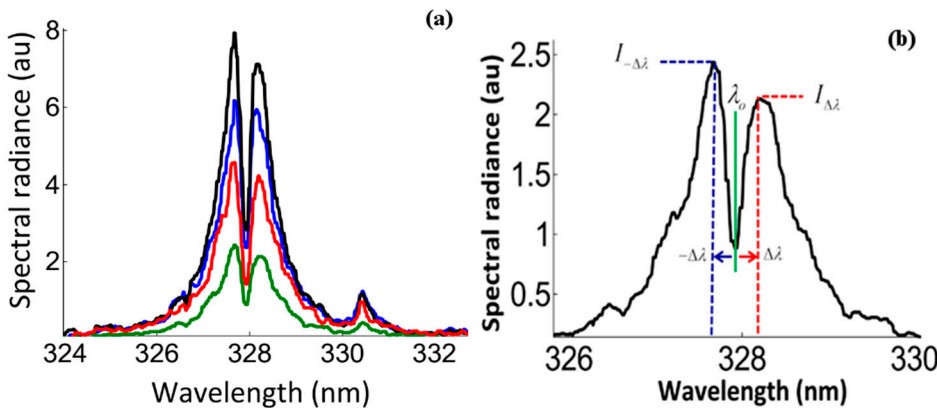

**Figure 5.** (**a**) Spectral radiance of the 327.9-nm line for different laser fluence. (**b**) Details for measurement of the spectral asymmetry.

The observed spectral line asymmetry, *As*, that appears for the 327.9-nm line can be calculated with $As = 2\left(\frac{I_{\Delta\lambda} - I_{-\Delta\lambda}}{I_{\Delta\lambda} + I_{-\Delta\lambda}}\right) \approx \left(\frac{hc}{\lambda_o k T_R} - 6\right)\frac{2\Delta\lambda}{\lambda_o}$ [4]. In this case, $I_{\Delta\lambda}$ and $I_{-\Delta\lambda}$ are the peak spectral radiances at red and blue wings, which is shown in Figure 5b. $T_R$ denotes the radiation temperature not the electron temperature $T_e$ because the laser-induced plasma is in local thermodynamic equilibrium (LTE), which implies that these temperatures are not equal i.e., $T_R \neq T_e$. Table 2 shows the amount of line asymmetry for decreasing laser fluence as well as values for the self-absorption factor, *SA*, and the Keldysh parameter, $\gamma$. The Keldysh parameter [32] is a measurement for the excitation mechanism, the $\gamma \ll 1$ and $\gamma \gg 1$ imply tunnel, and multiphoton ionization, respectively. Table 2 shows that multiphoton ionization generates laser-induced plasma in spite of large electromagnetic fields that may occur near nanoparticles.

**Table 2.** Line asymmetries, self-absorption factors, and Keldysh parameters for different fluence.

| Fluence (J/cm$^2$) | 13.4 | 10.9 | 6.9 | 5.6 | 4.8 | 3.9 | 2.1 |
|---|---|---|---|---|---|---|---|
| Line Asymmetry | 0.04 | 0.057 | 0.12 | 0.14 | 0.071 | 0.087 | 0.11 |
| Self-absorption, *SA* | 0.01 | 0.011 | 0.007 | 0.0048 | 0.0044 | 0.0022 | 0.0021 |
| Keldysh parameter, $\gamma$ | 470 | 520 | 660 | 730 | 790 | 880 | 1200 |

## 5. Discussion

Recent work elaborates on the interaction of the Nd: YAG radiation at wavelengths of 1064 nm, 532 nm, and 355 nm with silver nano-based targets and for different laser fluence levels in the range of 2 J/cm$^2$ to 13 J/cm$^2$. Three observations are identified for the resonance lines at 327.9 nm and 338.2 nm. Investigations of the spectral lines shapes reveal that:

(I)　Self-reversal is characterized by a large dip at the central wavelength [33],
(II)　Self-absorption, and
(III)　Asymmetries.

In the previously reported investigations, there was minimal if any experimentally recognizable trends of line asymmetry variations with either laser fluence or electron density. Yet for neutral emitters, asymmetric line profiles due to the Stark effect are predicted [4]. However, for further explanations of the observed phenomena in future work, one should evaluate effects associated with internally generated micro electromagnetic fields [4–7,24,33] in nanomaterial.

In addition, the spectral lines that arise from the Ag I at wavelengths of 768.7 nm and 827.35 nm are found to be optically thin. Consistent results are found when comparing electron densities to

those obtained from $H_\alpha$. Consequently, these two Ag I lines can be used as a standard spectral line to determine the plasma electron density when $H_\alpha$ is absent in the measured spectrum.

## 6. Conclusions

The emitted resonance spectral lines in nano-enhanced laser-induced plasma spectroscopy indicate self-reversal and asymmetries. Internal nanomaterial electromagnetic fields in the plasma may affect the observed line asymmetry. Further experimental and theoretical efforts are recommended for the explanation of the spectral line shapes from the nanomaterial plasma. Several lines at near-IR wavelengths are optically thin and these lines can be used as a reliable indicator of plasma electron density.

**Author Contributions:** A.M.EL.S. designed and performed the experiments; A.M.EL.S. and A.E.EL.S. analyzed the result together with C.G.P. and all three authors contributed to the writing of the article.

**Conflicts of Interest:** The authors declare no conflict of interest.

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
