# Peer review of "Measurement of Electron Density from Stark-Broadened Spectral Lines Appearing in Silver Nanomaterial Plasma"

_atoms, doi:10.3390/atoms6030044_

Round 1

Reviewer 1 Report

The authors of the paper

Measurement of electron density from Stark-broadened nanomaterial plasma

give some interesting consideration about laser induced plasma applied to the nanomaterial with silver nanoparticles. They described and analyzed the shapes of certain spectral lines and possibility to use some of the lines for plasma electron density determination.

In my opinion it is necessary to make some small corrections and improvement of the paper.

In fact, I recommend this paper for publication in Atoms after these minor revisions:

1. First, comment about the title. In the title is written “ …… Stark-broadened nanomaterial plasma”.  Stark broadening is effect that concerns of spectral lines, not nanomaterial plasma.

My suggestion for title is:

“Measurement of electron density from Stark-broadened spectral lines appearing in silver nanomaterial plasma”

or something similar.

2. General remark about references. In the text, the citation of the references appears in numerical order up to [22]. After that in page 4 (line 150) reference [29] is mentioned. At page 6 (line 182), after [24] again is mentioned reference [29].  Reference [28] is mentioned only on page 8 (line 241).

So, conclusion is, references should be given in numerical order.

3. The titles under the pictures 2, 4 and 5 and the title above Table 2 should be written with the reduced letters.

4. Please, write what is abbreviation TEM, page 3 (line 103).

5. In page 5 (line 166) at the end of the row is written “from e optically”. I suppose it should be “from the optically”.

6. It is not clear why the authors, separation between the peaks of the self-reversed lines (DlS2, line 234, Figs. 5 and 6), called FWHM (full width at half maximum)?

7. Why the authors used Halpha line as reference for electron density determination? Hbeta line is more appropriate for that purpose. Namely, the uncertainties for electron density determination by using Hbeta line are much smaller.

8. I suppose that electron density by using Halpha line is derived in conjunction with Griem’s or some other similar theory, but it is not mentioned in the text (page 5).

Which theory is used for electron density determination by using Ag I lines (Table 1and Table 2)?

The same question for numerical data in lines 242 and 243.

What are estimated errors? 

9. The electron densities obtained from Ag I 827 nm line, given in Tables 1 and 2, are not consistent.

It can be seen from the graph given below. Please, give comment about that.

I just realized that this graph can not be seen in this report. May be it can be seen

after printing?

10. Please, give some comment or definition about Keldysh parameter and also comment about

meaning the numbers given in Table 2. Why is this parameter important for the results obtained in this experiment? 

Author Response

We appreciate the diligence of reviewer 1, and agree with the valuable comments. The changes are indicated in green. Specifically, (1) modified the title as suggested, (2) renumbered the references, (3) reduced the font size for the figures and tables; (4) spelled out TEM; (5) corrected typos; (6) removed the section from original lines 229-252 (in response to reviewer 2), therefore, the original Figs. 5 and 6 are removed; (7) included a comment RE Hβ, (8) included references for Hα and Ag I lines, and an error estimate, (9) Table 2 showed larger ne due to issues with self-absorption, but with removal of pages 229-252, the ne line is no longer needed and would confuse things, and (10) included a reference and text regarding the Keldysh parameter.

Reviewer 2 Report

The authors have demonstrated deep understanding of the research topic and clearly presented the methods, results and draw the conclusions.

I suggest this manuscript to be accepted for publication in Atoms journal after the following updates:

On the line 42, spectral intensity is given in Watt/ m3 sr. Please correct the units to W⋅sr−1⋅m−1, or use the intended radiometric units if the spectral intensity is improperly selected.

On the lines 287-290, the authors are calling the references in which is stated that there was no recognizable trend of the line asymmetry of the measured Ag I line profiles. This should be taken cautiously because it is known that the Stark effect induced line profiles of neutral emitters are asymmetric (Ref. 4.) . Consequently, the authors are restoring the line profiles of Ag I 328 and 338 nm to symmetric profile, even if the measured profiles are asymmetric. The asymmetry of Ag I 328 and 338 nm lines is not negligible as it is for Ag I 769 and 827 nm profiles. This is documented in Figs. 2-7.

The corrected FWHM values for Ag I 328 and 338 nm reported in Fig. 6 are about 0.15 to 0.2 nm. To get these FWHM values from the Stark broadening calculations in Semiclassical formalism of Dimitrijevic and Sahal-Brechot,(DSB) the plasma electron concentration should be in 3-4 x 1E18 cm^-3 and the electron temperatures in 10000 K ranges (follow the link: https://stark-b.obspm.fr/index.php/table and select Ag I 328 and 338 nm transitions). These plasma concentrations are similar to those reported on Line 242 for the unrestored line profiles. The electron concentration measured by using the optically thin line at 827.4 nm for the same experimental conditions is significantly lower (2.9E17 cm^-3). This implies that either: a) your plasma electron concentration is an order of magnitude higher in the center of your plasma plume, where Ag I 328 and 338 nm radiation originate, compared to the plume periphery where Ag I 769 and 827 nm radiate, or b) if your plasma electron concentration is order of 1E17 cm^-3 and below, then your line profile restoration requires use of the function of asymmetric profile (Ref. 4) which will not likely help you to restore the line profile's FWHM to match the 1E17 cm^-3 plasma electron concentrations. Consequently, option a) is the one that applies to your plasma conditions.

For this reason, my first recommendation is to remove the analysis of Ag I 328 and 338 nm profiles before the manuscript can be accepted for publication.

Alternatively, you may modify the method to restore these line profiles to match higher plasma electron concentrations, but this will require major rework.

Author Response

We are very thankful for the valuable and accurate comments from reviewer 2, and modified the manuscript as follows in red: original line 42 should show spectral radiance – unfortunately, frequently “spectral intensity” is confusingly used. (However, we leave the ordinate labels “spectral intensity (au)”.) The original lines 229-252 are removed in the manuscript, we fully agree with the comment (and perhaps defer further analysis to future work).